# Polycomb- and REST-associated histone deacetylases are independent pathways toward a mature neuronal phenotype

James C McGann[1,2][*][†], Jon A Oyer[1,2][†][‡], Saurabh Garg[1,2], Huilan Yao[1,2], Jun Liu[1,2][§], Xin Feng[3], Lujian Liao[4][¶], John R Yates III[4], Gail Mandel[1,2]

[1]Vollum Institute, Oregon Health and Science University, Portland, United States; [2]Howard Hughes Medical Institute, Chevy Chase, United States; [3]Department of Molecular and Human Genetics, Baylor College of Medicine, Houston, United States; [4]Department of Chemical Physiology, Scripps Research Institute, La Jolla, United States

*For correspondence: mcgann@ohsu.edu

[†]These authors contributed equally to this work

Present address: [‡]Division of Hematology/Oncology, Robert H Lurie Comprehensive Cancer Center, Feinberg School Of Medicine, Chicago, United States; [§]Department of Biology, Indiana University, Bloomington, United States; [¶]Shanghai Key Laboratory of Regulatory Biology, School of Life Sciences, East China Normal University, Shanghai, China

Competing interests: The authors declare that no competing interests exist.

**Abstract** The bivalent hypothesis posits that genes encoding developmental regulators required for early lineage decisions are poised in stem/progenitor cells by the balance between a repressor histone modification (H3K27me3), mediated by the Polycomb Repressor Complex 2 (PRC2), and an activator modification (H3K4me3). In this study, we test whether this mechanism applies equally to genes that are not required until terminal differentiation. We focus on the RE1 Silencing Transcription Factor (REST) because it is expressed highly in stem cells and is an established global repressor of terminal neuronal genes. Elucidation of the REST complex, and comparison of chromatin marks and gene expression levels in control and REST-deficient stem cells, shows that REST target genes are poised by a mechanism independent of Polycomb, even at promoters which bear the H3K27me3 mark. Specifically, genes under REST control are actively repressed in stem cells by a balance of the H3K4me3 mark and a repressor complex that relies on histone deacetylase activity. Thus, chromatin distinctions between pro-neural and terminal neuronal genes are established at the embryonic stem cell stage by two parallel, but distinct, repressor pathways.

## Introduction

Undifferentiated pluripotent cells present a unique dilemma with regard to gene regulation; genes that promote differentiation must be repressed to maintain pluripotency, yet this repression must be reversible to allow for rapid response to developmental cues. The repressed status, often referred to as poised, is conferred by epigenetic modifications established at loci encoding developmental regulators. Specifically, global histone modification patterns in embryonic stem cells (ESCs) have revealed the coexistence of trimethylation of histone H3 at lysines 4 and 27 (H3K4me3 and H3K27me3) at promoters of genes encoding key lineage-determining factors (*Bernstein et al., 2006*). This dual chromatin status has been termed bivalence to reflect the juxtaposition of modifications typically associated with functionally active and transcriptionally repressed promoters, respectively. The H3K27me3 mark is established by the methyltransferase EZH2 within the Polycomb Repressor Complex 2 (PRC2) (*Pengelly et al., 2013*), which effectively provides a counterbalance to factors that promote H3K4me3 and active expression (*Bernstein et al., 2006*). This PRC2-dependent state has been proposed as a universal mechanism to confer pluripotency by controlling all developmental lineages, but its application to the neuronal lineage has not been tested rigorously. This is of interest because for most lineages the key developmental regulators are activators, while in the neuronal lineage a key regulator is the transcriptional repressor REST (NRSF).

Specifically, REST is a master developmental regulator that controls a large suite of genes that encode proteins critical for neuronal maturation, such as cellular migration, axonal pathfinding,

**eLife digest** When an embryo is developing, genes are switched on or off at different times, for many different reasons. Many of these genes are switched off, or repressed, by making the DNA inaccessible to the various proteins and molecules that control gene activity. This is achieved by altering the way that the DNA is packaged into a compacted structure called chromatin. A host of proteins modify the structure of chromatin: it can be made more tightly packaged, which keeps genes switched off; or it can be made more loosely packaged, which allows the genes within to be accessed and switched on.

The stem cells in an embryo are able to give rise to many different types of specialized cell. Genes that determine which cell type a stem cell will eventually become are often kept in a so-called 'poised' state, and have chromatin modifications that encourage genes to be switch on, as well as modifications that switch genes off. Current thinking is that this poised state allows these genes to be switched on or off rapidly in response to the signals that the cell receives during development.

The only known protein complex that causes the chromatin to become more compacted in this poised state is the Polycomb complex. This complex binds to specific regions of DNA and is thought to allow stem cells to remain able to become different cell types by repressing the genes required for adopting a specialized cell fate. However, it is unclear if this poised state also regulates those genes that control the final stages of a cell becoming a specific cell type.

McGann et al. investigated genes that are involved in the final stages of a nerve cell's development. These genes are regulated by another protein called REST, which acts to repress the genes in non-neuronal cells. McGann et al. found that the genes that are regulated by REST in embryonic stem cells from mice also have their chromatin modified in two contrasting ways. Some of the modifications are linked to switching genes on, while others are linked to keeping genes switched off. Thus these genes are also in a poised state. However, for these genes, this state is acquired without the activity of the Polycomb complex.

The results of McGann et al. show that two similar, but distinct, mechanisms keep the genes required for the early and the late stages of nerve cell development in a poised state. If this poised state aids the development of other cell types (for example muscle or fat cells), uncovering how it is achieved could improve our ability to direct stem cells to develop into specific cell types and tissues.

and synaptic transmission (**Johnson et al., 2007**, **2008**; **Otto et al., 2007**). Further, REST is expressed at very high levels in embryonic stem cells, contrary to other developmental regulators. A global REST knockout results in embryonic lethality, pointing to an essential function for REST in general embryonic development following the ESC stage (**Chong et al., 1995**; **Schoenherr and Anderson, 1995**). In neural progenitors, REST levels decrease until it completely leaves the chromatin at terminal differentiation of most neurons. Preventing its dismissal from chromatin delays greatly neuronal maturation in vivo (**Mandel et al., 2011**) and alters neural progenitor pool identities in vitro (**Covey et al., 2012**). In stem/progenitor cells, developmental genes required for neuronal lineage decisions are repressed, including pro-neural and REST-regulated genes (**Buckley et al., 2009**, **Ballas et al., 2005**), but whether the mechanisms that regulate these classes of genes are the same or different remains an open question.

Prior studies have shown an important role for PRC2 repression on poised genes of multiple lineages. On the one hand, although the terminal neuronal genes regulated by REST are poised in stem cells, REST is itself a repressor and may not require the additional repression mechanism of Polycomb. On the other hand, the non-coding RNA (ncRNA) HOTAIR has been shown to act as an adapter between the core PRC2 component EZH2 and the REST co-factor Kdm1a (**Tsai et al., 2010**), suggesting a connection between PRC2 and REST in ESCs. In addition, other groups have observed biochemical interaction between REST and PRC2 members (**Dietrich et al., 2012**; **Mozzetta et al., 2014**) and recruitment of H2K27me3 to RE1 sites (**Arnold et al., 2013**). Therefore, we performed three studies to test directly for the existence of a functional relationship between PRC2 and REST in ESCs. First, we performed a mass spectrometric analysis of REST complexes to identify ESC-specific co-factors in an unbiased manner. Second, we asked whether REST-occupied neuronal genes were marked by

H3K27me3, and furthermore, whether PRC2 activity was compromised in *Rest*<sup>−/−</sup> ESCs. Finally, exploiting a *Rest*<sup>−/−</sup> ESC line, we examined the consequences of the loss of REST on chromatin marks and gene expression.

## Results

### REST complexes purified from ESCs

Previous studies of REST-interacting proteins in ESCs used a candidate approach and focused on co-factors characterized in differentiated cells (*Ballas et al., 2005*; *Yu et al., 2011*). In the current study, we considered the possibility that ESC-specific co-factors might be involved in regulatory mechanisms of REST that were unique to pluripotent cells. To test this idea we performed a mass spectrometric analysis of REST complexes using a mouse ESC line that stably expressed both the biotin conjugating enzyme, BirA (*Kim et al., 2009*), and REST tagged with a biotin acceptor sequence. The stable line expressed approximately five-fold higher levels of REST than normal ESCs with no differences in pluripotency markers compared to WT cells (not shown). Multidimensional Protein Identification Technology (MudPIT) analysis was performed on three independent streptavidin purifications. Proteins that were co-purified with REST in at least two of three pull-downs and were weakly represented, if at all, in the BirA control pull-downs are shown in *Table 1*. None of the known epigenetic regulators identified as co-factors by mass spectrometry were specific to ESCs. However, we did identify almost all known REST co-factors including CoREST1 and Sin3a as well as the chromatin modifying enzymes, HDAC1 and 2, Kdm1a and G9a/Glp, and the G9a-associated adaptors CDYL and WIZ1, all of which have been shown biochemically to be present within REST complexes in terminally differentiated cell types (*Andres et al., 1999*; *Grimes et al., 2000*; *Hakimi et al., 2002*; *Roopra et al., 2004*; *Mulligan et al., 2008*), thus validating our approach. We noted that an additional CoREST family member, CoREST2, was also present in the pull-downs. We confirmed the presence of CoREST2, as well as a subset of other co-factors, at RE1 sites in ESCs by chromatin immunoprecipitation (ChIP, *Figure 1—figure supplement 1*). We also identified several new factors, some with known functions (Smarca5, Mdc1) and some with no known function (D1Pas1, *Table 1*). In contrast to these factors, components of the Polycomb repressor complexes were not identified according to our criteria. It was possible that the specific conditions used to generate the whole-cell extracts used in the MudPIT analysis precluded identification of Polycomb proteins. Therefore, we repeated mass spectrometry analysis on streptavidin pull-downs from nuclear extracts (*Abmayr et al., 2006*). Under these conditions, we did identify the PRC2 complex members Suz12 (3 and 4 peptides in BioT REST pull-down replicates, 0 and 0 peptides in Control) and Ezh2 (3 and 5 peptides in BioT REST, 0 and 0 peptides in Control). Co-immunoprecipitation analysis using nuclear extract confirmed only the Suz12 interaction, as well as the interactions with known REST co-repressors (*Figure 1—figure supplement 2A*). Importantly, however, the members of the PRC2 complex required for the methyltransferase activity, Ezh2, and for complex formation, Eed (*Montgomery et al., 2005*), were both absent from the co-immunoprecipitation (*Figure 1—figure supplement 2A*). These results indicate that REST protein does not interact with an enzymatically active PRC2 complex in ESCs. To supplement this proteomic approach, and as an independent test for the role of PRC2 members in REST regulation, we used a genome-wide ChIP-seq approach.

### The majority of REST-occupied sites, including promoters, are regulated independently of PRC2

A Polycomb complex was not represented in our analysis of REST complexes, but it was possible that the streptavidin pull-downs might not co-purify ncRNA-mediated associations. Therefore, we compared the genomic distributions of REST and H3K27me3 enrichment to determine whether PRC2 is recruited to REST-bound sites in ESCs. 2136 genomic regions targeted by REST were identified by our ChIP-seq in mouse ESCs, which is comparable to prior REST ChIP-seq studies in human T cells (*Johnson et al., 2007*) and a different mouse ESC line (*Johnson et al., 2008*). The DNA sequence of REST-bound regions was analyzed and 96.6% (2064 REST-bound sites) contained either the complete or partial consensus RE1 sequence motif (*Otto et al., 2007*) (*Figure 1—figure supplement 3B*). This sequence analysis also showed that the RE1 sequence is the central determinant for REST recruitment because regions with the highest enrichment contained multiple repeats of complete RE1 sites aligned with the same strand orientation (*Zhang et al., 2006*; *Jothi et al., 2008*) (*Figure 1—figure supplement 3A*). Conversely, regions that contained a single right half of the RE1 motif were associated

**Table 1.** Co-factors identified within REST complexes were purified from ESCs

| Functional category | Gene | Experiment 1 | | Experiment 2 | | Experiment 3 | |
|---|---|---|---|---|---|---|---|
| | | BioT REST | Control | BioT REST | Control | BioT REST | Control |
| Bait | REST | 41 | | 37 | | 27 | |
| Corepressor | Rcori | 4 | | 8 | | 4 | |
| | Rcor2 | 8 | 2 | 19 | 3 | 8 | |
| | Sin3a | 8 | | 11 | | 6 | |
| Histone tail modifying enzyme | HDAC1 | 11 | | 12 | | 7 | 3 |
| | HDAC2 | 7 | | 9 | 2 | 4 | |
| | LSD1 | 18 | | 26 | | 21 | 3 |
| | Prmt5 | 2 | | 2 | | 2 | |
| | Wdr5 | 3 | | 2 | | | |
| | Ehmt2/G9a | 5 | | 18 | | 8 | |
| | Ehmti | | | 4 | | 6 | |
| | Wiz | | | 4 | | 6 | |
| Adaptor | Cdyl | 5 | | 5 | | 3 | |
| | Cdyl2 | 3 | | 3 | | | |
| Chromatin remodeler | Smarca5 | 3 | | 4 | | 5 | 2 |
| | Supt16h | 3 | | 4 | 4 | 6 | |
| | Ssrpi | 3 | | | | 2 | |
| Other repressor | Gata2b | 3 | | | | 2 | |
| | MBD3 | | | 2 | | 4 | |
| F-box protein | Fbxwi 1 | 2 | | 9 | | 4 | |
| | Btrc | 2 | | 6 | | | |
| Transposase | Lin28A | | | 4 | | 2 | |
| | Trim71 | 2 | | 2 | | | |
| DNA binding | Mdd | 2 | | | | 2 | |
| | Bclafi | 2 | | | | 2 | |
| | Utf1 | 2 | | | | 3 | |
| Unclear | Bxdc2 | 2 | | 2 | | 3 | |
| | D1Pas1 | 8 | | 4 | | 11 | |
| | Gcdh | 3 | | 6 | | 2 | |
| | Pdcd11 | 2 | | 4 | | 2 | |
| | Pop1 | 2 | | 2 | | 2 | |
| | Wwox | 2 | | 4 | | | |
| | Pura | | | 3 | | 2 | |
| | Dimti | 2 | | 2 | | | |

Proteins are listed were identified in all streptavidin purifications of biotin-tagged REST (3 out of 3) but not represented in more than one of the negative control samples. Columns list the functional category, protein symbol, and the number of unique peptides detected in REST and negative control purifications.

**Source data 1**. REST-bound genomic regions with repeated consensus RE1 motifs. Columns list the chromosome and base pair coordinates (Region Start & Region End) of the REST-binding domain identified by PeakRanger analysis of ChIP-Seq read distribution. RE1 Start and RE1 End columns give the coordinates corresponding to the positions of individual RE1 motifs found by FIMO within the corresponding region. Orientation column lists whether the RE1 motif is on the forward (+) or reverse (−) DNA strand, and the p-value column gives the calculated log-odds score from the comparison of a discovered motif to a position weighted matrix corresponding to the full consensus RE1 motif.

with low levels of enrichment (*Otto et al., 2007*) (*Figure 1—figure supplement 3A*). REST-binding and relative enrichment indicated by ChIP-seq were confirmed by ChIP-quantitative PCR analysis for a subset of loci (*Figure 1—figure supplement 3C*). Although ncRNA-mediated interactions linking REST to PRC2-bound chromatin have been proposed (*Tsai et al., 2010*), the strong correlation between REST-binding and RE1 DNA sequences suggests that any alternative mechanisms of stable recruitment to chromatin were not prevalent in ESCs. This did not preclude, however, the inverse possibility that REST could recruit PRC2 to chromatin adjacent to RE1 sites. However, assessment of H3K27me3 domains showed that only a small minority (~3%) of REST-bound sites were associated with significant enrichment of H3K27me3 relative to input, and only 0.5% of H3K27me3-enriched domains were associated with REST binding (*Figure 1A*). Even if the effective footprint of REST sites was extended 1 kb in both directions, the proportion overlapping with H3K27me3 peaks was only 12.6% of REST peaks and 2.3% of H3K27me3 peaks. Furthermore, our own analysis of ChIP-seq results published previously show that the PRC2 factors Suz12 and Ezh2 bind at extremely low levels, if at all, at REST sites relative to sites of H3K27me3 enrichment (*Figure 1—figure supplement 2B*). None of these data sets supports a strong functional connection between these distinct complexes, or between RE1 sites and PRC2, in ESCs.

Despite the limited association between H3K27me3 enrichment and REST binding, we asked whether promoters targeted by both repressive mechanisms (REST and PRC2) represented specific gene classes, because H3K27me3 marks several key developmental factors in ESCs. The dual PRC2/REST-occupied genes were primarily canonical REST-regulated mature neuronal genes rather than pro-neural or developmental genes per se, and therefore showed no ontological category enrichment (data not shown). Developmental regulators of multiple lineages, such as Cdx4 and Runx1, which are associated with the bivalent marks H3K27me3 and H3K4me3 (*Mikkelsen et al., 2007*), similarly showed no gene ontology differences between those occupied by REST and those that were not. These results suggest that there exists no specific functional class of genes that is regulated by REST and Polycomb in tandem.

To determine whether PRC2 activity at REST-occupied sites, when it did occur, was dependent on REST binding, we asked whether H3K27me3 was lost from these regions in *Rest*−/− ESCs. We integrated the ChIP-seq signal across both narrow REST-binding domains and across a continuous broad domain to avoid nucleosome occupancy fluctuations due to loss of REST binding. By this analysis, we found that levels of H3K27me3 at defined H3K27me3 sites were largely maintained (*Figure 1B*, Pearson's coefficient = 0.53). The changes observed between WT and *Rest*−/− ESCs were very similar to those observed between WT data sets published previously (*Figure 1—figure supplement 4*). Specifically, we found that >95% of H3K27me3-enriched regions associated with REST-bound sites in WT cells were also enriched for H3K27me3 in *Rest*−/− ESCs (*Figure 1A* and C, Pearson's coefficient = 0.85). The number of H3K27me3-overlapping domains was also essentially the same between wild-type and REST knockout ESCs (*Figure 1A*). Similar results were observed when only REST sites within 5 kb of gene promoters were analyzed (data not shown). In the small number of instances where H3K27me3 levels did change, some genes lost H3K27me3 in *Rest*−/− ESCs, including Scn8a, Galnt9, and Vgf (*Figure 1D* and *Table 2*), while other genes, including Dner, Otop3, and Cacng2, gained H3K27me3 (*Figure 1D* and *Table 2*). The losses and gains were validated by quantitative ChIP-PCR (*Figure 1D*). The HoxA11 and Oct4 genes, which are not bound by REST in ESCs, represent positive and negative controls for the H3K27me3 mark, respectively. The occupancy of EZH2 at these same regions was altered similar to H3K27me3 (*Figure 1—figure supplement 5*). These results taken together indicated that REST was not required for establishment or maintenance of H3K27me3, throughout the genome generally or at loci targeted by REST specifically. However, in a very limited number of cases, H3K27me3 was lost in *Rest*−/− ESCs, reflecting either direct or indirect influence of REST on PRC2-binding. The increase in H3K27me3 at select sites may reflect block of PRC2 binding by REST due to close proximity of their binding sites. Why certain dually occupied genes lost or gained H3K27me3 in response to loss of REST was not obvious based on the function of the encoded proteins, but could be related to the timing of expression in vivo.

## Chromatin marks at the REST-binding site provide a signature for neuronal genes in ESCs

Although the above experiments ruled out a major role for PRC2 in REST-regulated repression, our mass spectrometry results using whole-cell extracts revealed three histone modifying enzymes in the

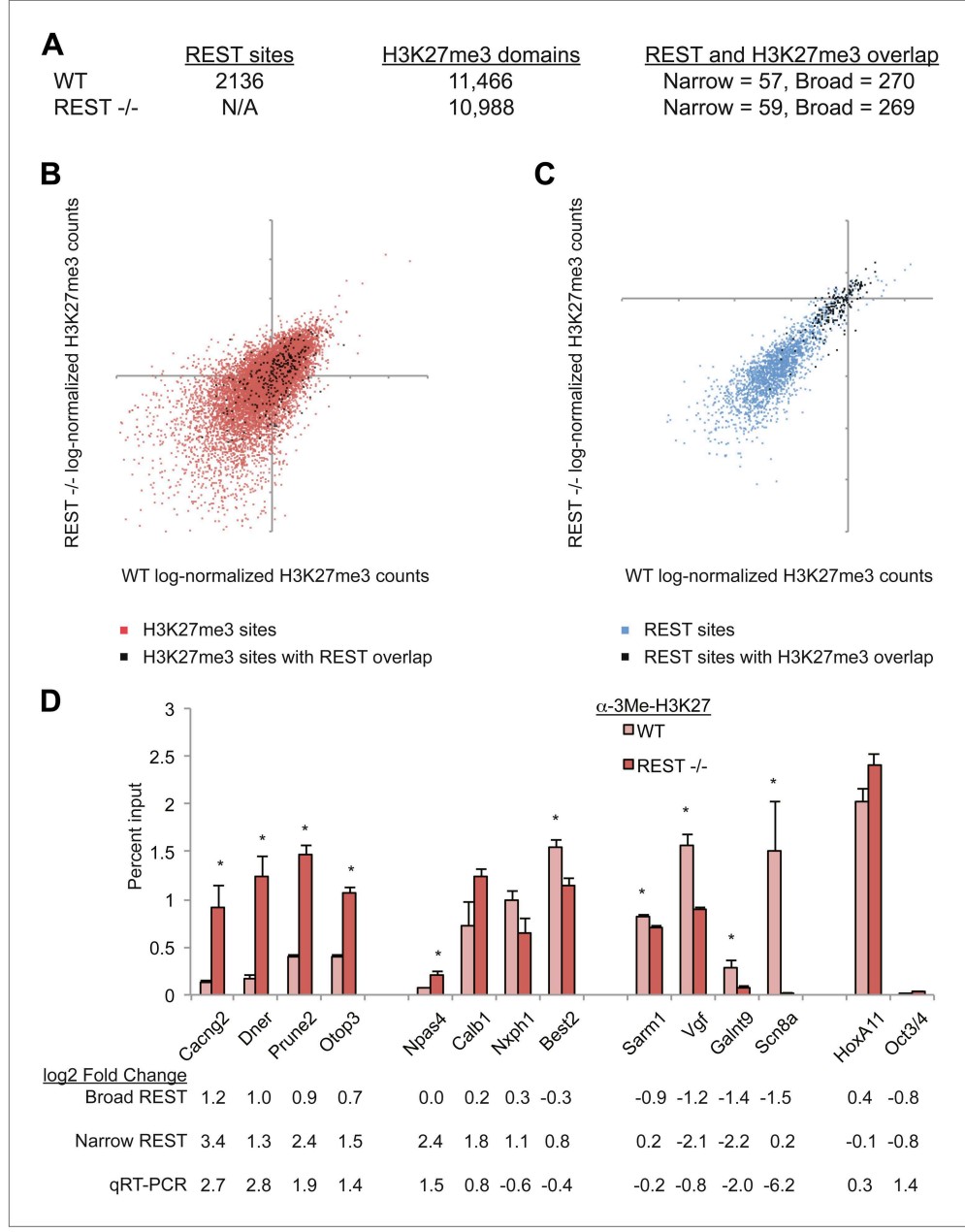

**Figure 1**. PRC2 establishes H3K27me3 in ESCs independent of REST repression. (**A**) A limited number of REST-occupied sites are associated with domains of H3K27me3 enrichment in ESCs, even if defined more broadly (+/− 1 kb). (**B**) H3K27me3 levels are stable in *Rest*⁻/⁻ ESCs in the majority of regions targeted by PRC2. The scatter-plot shows the relative enrichment of H3K27me3 ChIP-Seq signal in wild type (WT, x-axis) and *Rest*⁻/⁻ ESCs (y-axis) at regions targeted by PRC2 in WT ESCs. (**C**) As in (**B**), but at identified REST-binding sites. (**D**) Chromatin immunoprecipitation analysis showing H3K27me3-enrichment changes at RE1 sites near PRC2-targeted regions in WT and *Rest*⁻/⁻ ESCs (* indicates $p < 0.05$), normalized for H3 density.

The following figure supplements are available for figure 1:

**Figure Supplement 1**. REST is required for recruitment of co-factors to RE1 sites in ESCs.

**Figure Supplement 2**. Detection of REST binding to PRC2 members is biochemically possible, but a true interaction is unlikely.

*Figure 1. Continued*

**Figure Supplement 3**. Characteristics of REST-bound loci.

**Figure Supplement 4**. H3K27me3 levels from WT and *Rest*[−/−] ESCs are as similar as H3K27me3 levels from different published reports.

**Figure Supplement 5**. Ezh2-enrichment at REST-bound loci.

purified REST complex that are often associated with repression: the histone H3K9 methyltransferase, G9a, histone deacetylases (HDACs) 1 and 2, and the histone H3K4me1/2 demethylase, Kdm1a. Chromatin immunoprecipitation analysis showed that G9a recruitment was lost at RE1 sites in *Rest*[−/−] ESCs (***Figure 1—figure supplement 1***), and we observed a significant reduction in the levels of H3K9me2 in the region of the RE1 sites in 12/16 genes (***Figure 2A***). Two control genes expressed in ESCs but lacking RE1s did not show any change (***Figure 2A***). Consistent with the above findings, there was no correlation between changes in H3K9me2 in WT and *Rest*[−/−] ESCs and changes in H3K27me3 ($R^2 = 0.002$), again underscoring the independence of REST and PRC2 in chromatin remodeling.

**Table 2.** REST-associated genes with significant changes in H3K27me3 levels were measured in *Rest−/−* ESCs

| Gene Symbol | Gene Name | Change in *Rest*[−/−] ESC |
|---|---|---|
| Prune2 | Prune homolog 2 | 2.1 |
| Fosb | FBJ osteosarcoma oncogene B | 2.0 |
| Mast1 | Microtubule associated serine/threonine kinase 1 | 1.8 |
| Celf4 | Bruno-like 4, RNA binding protein | 1.7 |
| Kiaa1152 | Uncharacterized protein C14orf118 homolog | 1.7 |
| Dner | Delta/notch-like EGF-related receptor | 1.6 |
| Cacng2 | Stargazin | 1.6 |
| Bdnf | Brain derived neurotrophic factor | 1.6 |
| Hes3 | Hairy and enhancer of split 3 | 1.6 |
| Otop3 | Otopetrin 3 | 1.5 |
| A330050F15Rik | Uncharacterized protein LOC320722 | 1.5 |
| Skor2 | SKI family transcriptional corepressor 2 | 1.4 |
| Nmnat2 | Nicotinamide nucleotide adenylyltransferase 2 | −1.3 |
| Cnnm1 | Cyclin M1 | −1.3 |
| Cabp1 | Calcium binding protein 1 | −1.5 |
| Kcnb1 | K⁺ voltage gated channel, Shab-related subfamily | −1.6 |
| Celsr3 | Flamingo homolog 1 | −1.7 |
| Mapt | Microtubule-associated protein tau isoform a | −1.7 |
| Bsn | Bassoon | −2.2 |
| Vgf | VGF nerve growth factor inducible | −4.4 |
| Sarm1 | Sterile alpha and TIR motif containing 1 | −12.1 |
| Galnt9 | Polypeptide Gal NAc transferase 9 | Loss |
| Smpd3 | Sphingomyelin phosphodiesterase 3 | Loss |
| Scn8a | Na$^{2+}$ voltage-gated channel, type VIM, alpha | Loss |

List of genes located near REST bound regions that were associated with PRC2 in WT ESCs and showed a significant difference in H3K27me3 enrichment relative to *Rest*[−/−] ESCs.

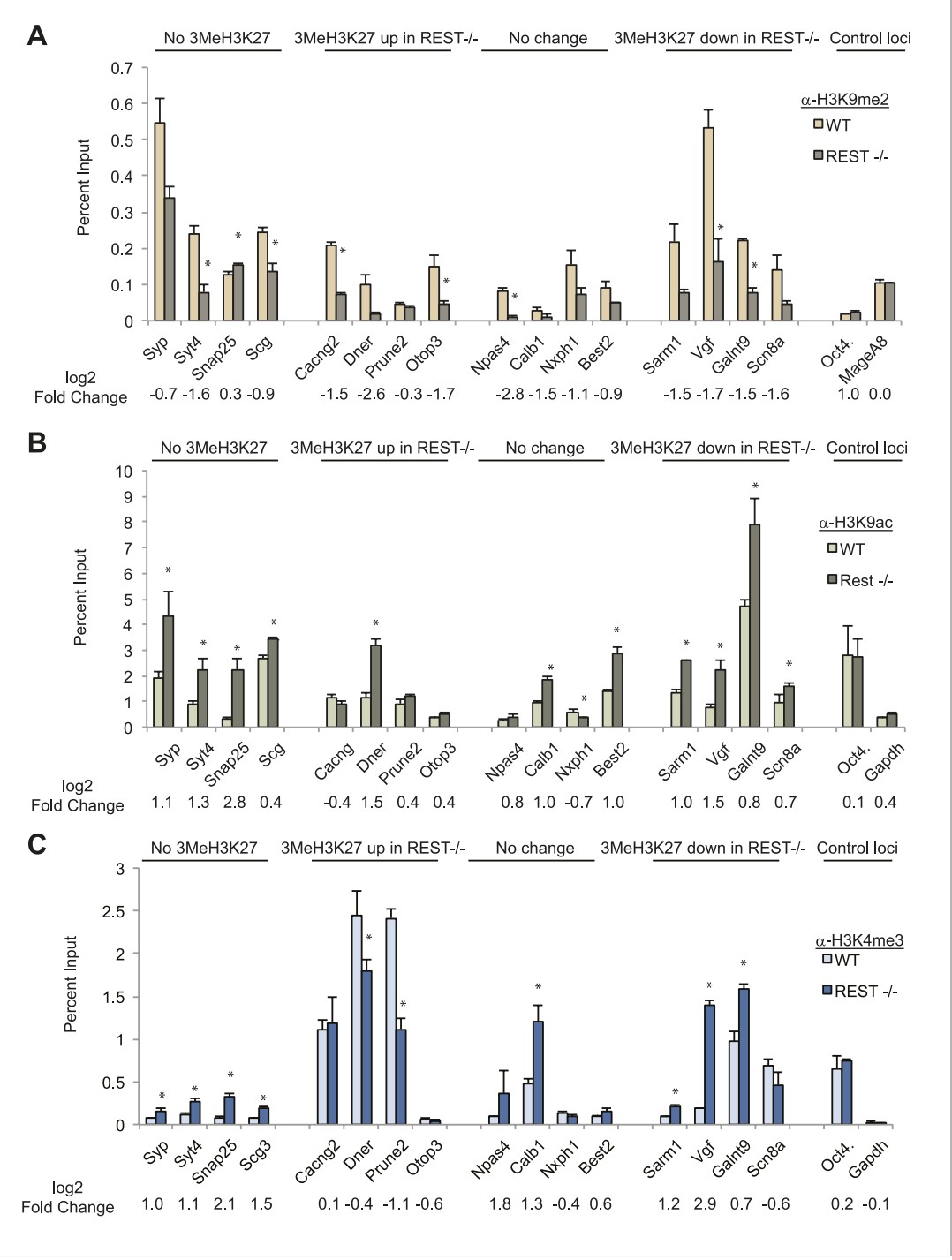

**Figure 2**. Chromatin modification changes due to loss of REST. (**A**) REST-dependent establishment of 2Me-H3K9, measured by ChIP, is impaired at RE1 sites in *Rest*$^{-/-}$ ESCs irrespective of changes in H3K27me3 levels. Oct4 and MageA8 are genes expressed in ESCs that lack RE1 sites. (**B**) Increased histone acetylation is detected at most RE1-associated promoters in the absence of REST, irrespective of changes in H3K27me3 levels. *Oct4* and *Gapdh* promoter regions are expressed in ESCs and lack RE1 sites. (**C**) H3K4me3 enrichment is increased at most RE1-associated promoters in *Rest*$^{-/-}$ ESCs, independent of H3K27me3 levels (* indicates $p < 0.05$).

The presence of HDACs in the REST complex predicted increased H3K9ac enrichment at RE1 sites in *Rest*[−/−] ESCs, which we observed in 11/16 analyzed genes, with no change in the controls that lacked RE1 sites (*Figure 2B*). Importantly, there was no correlation between enrichment of H3K9ac and gain or loss of H3K27me3 ($R^2$ value = 0.056) due to the loss of REST.

Although MLL proteins were not present in the REST immuno-complex, we tested for the presence of the H3K4me3 mark because it is associated with active or 'poised' promoters in ESCs in opposition to H3K27me3 (*Bernstein et al., 2006*) or REST (*Ballas et al., 2005*). Of 16 genes containing RE1 sites, H3K4me3 was increased significantly in eight of them in *Rest*[−/−] ESCs, independent of the presence of the H3K27me3 mark and whether it was altered by the loss of REST (*Figure 2C*; $R^2$ = 0.029). Thus, in the context of the bivalent hypothesis, although Polycomb is not an active component for REST-regulated genes, the presence or absence of the H3K4me3 mark may be an important aspect of the chromatin signature orchestrated by REST. As further evidence for this, we found that 37% (441/1202) of REST sites within 20 kb of target genes overlapped with H3K4me3 peaks, a number that increased to 62% for those REST sites within 5 kb of the TSS (417/617).

## REST-dependent H3K4me3 changes in ESCs coincide with changes in neuronal gene expression

To determine the functional consequences of chromatin modification changes due to the loss of REST repression, we performed RNA-seq on transcripts from WT and *Rest*[−/−] ESCs. As expected, numerous REST target genes (binding site identified within 20 kb of the transcription start site) show an expected increase in expression levels in *Rest*[−/−] ESCs. However, the expression data show no correlation with changes in H3K27me3 levels, either at the REST-binding site (*Figure 3A*) or at the TSS (*Figure 3B*). From this analysis we conclude that even the small H3K27me3 changes observed due to the loss of REST have little effect on the expression of REST target genes, further supporting the functional independence of REST from Polycomb. When REST target genes are further categorized according to their promoter status regarding H3K27me3 and H3K4me3 (*Young et al., 2011*) in WT ESCs, it is evident that all classes of REST target genes are de-repressed, irrespective of other marks (*Figure 3C*). In comparison, REST target genes are not de-repressed in *Eed*[−/−] ESCs (*Ferrari et al., 2014*) (*Figure 3D*), which show drastic decreases in the H3K27me3 mark. Combined, these results support the conclusion that REST is the primary repressor of its target genes and the roles of Polycomb and the H3K27me3 mark are functionally dispensable for its activity.

Bivalent developmental genes that become activated during differentiation are proposed to lose the repressive H3K27me3 mark but maintain the active H3K4me3 mark. Having shown that the REST and Polycomb pathways were largely independent, we asked which REST-dependent chromatin marks at the ESC stage might influence transcript levels of these targets. To this end, we performed a multiple regression analysis to determine which chromatin changes due to the loss of REST at the ESC stage were most likely associated with the observed expression changes. From this analysis, only the chromatin mark H3K4me3 correlated significantly with the expression changes observed in *Rest*[−/−] ESCs (p <0 0.02, *Figure 3E*). Further support for the importance of the H3K4me3 mark at REST targets is provided by the absolute levels of expression of REST target genes when categorized as above. Specifically, genes occupied by REST, marked either by just H3K4me3 or by H3K4me3 together with H3K27me3, show significantly lower expression levels than non-REST target genes (*Figure 3F*, p < 0.001 and p < 0.05, respectively). When REST is deleted from ESCs, the expression levels of these H3K4me3-marked REST target genes are increased and approaches that of non-target H3K4me3-marked genes. This REST-dependent repression of H3K4me3-enriched promoters suggests that one of the primary functions of REST in ESCs is to counter RNA Pol II recruitment and maintenance of this activating mark.

## REST antagonizes H3K4me3 through the activity of histone deacetylases

Despite functioning independently, Polycomb and REST repressor complexes in ESCs can generate similar downstream molecular effects by blunting H3K4me3 signaling at transcriptionally poised genes required for differentiation to proceed. To identify the mechanism for the increases in H3K4me3 after the loss of the REST complex, we monitored H3K4me3 levels at REST sites in ESCs that were mutant for the co-repressors G9a, Kdm1a, or HDACs (*Figure 4A*). We used the histone deacetylase inhibitor trichostatin-A (TSA) as a proxy for HDAC loss, due to the redundancy of HDAC family members (*Montgomery et al., 2007*). Only TSA treatment correlated significantly with increased H3K4me3

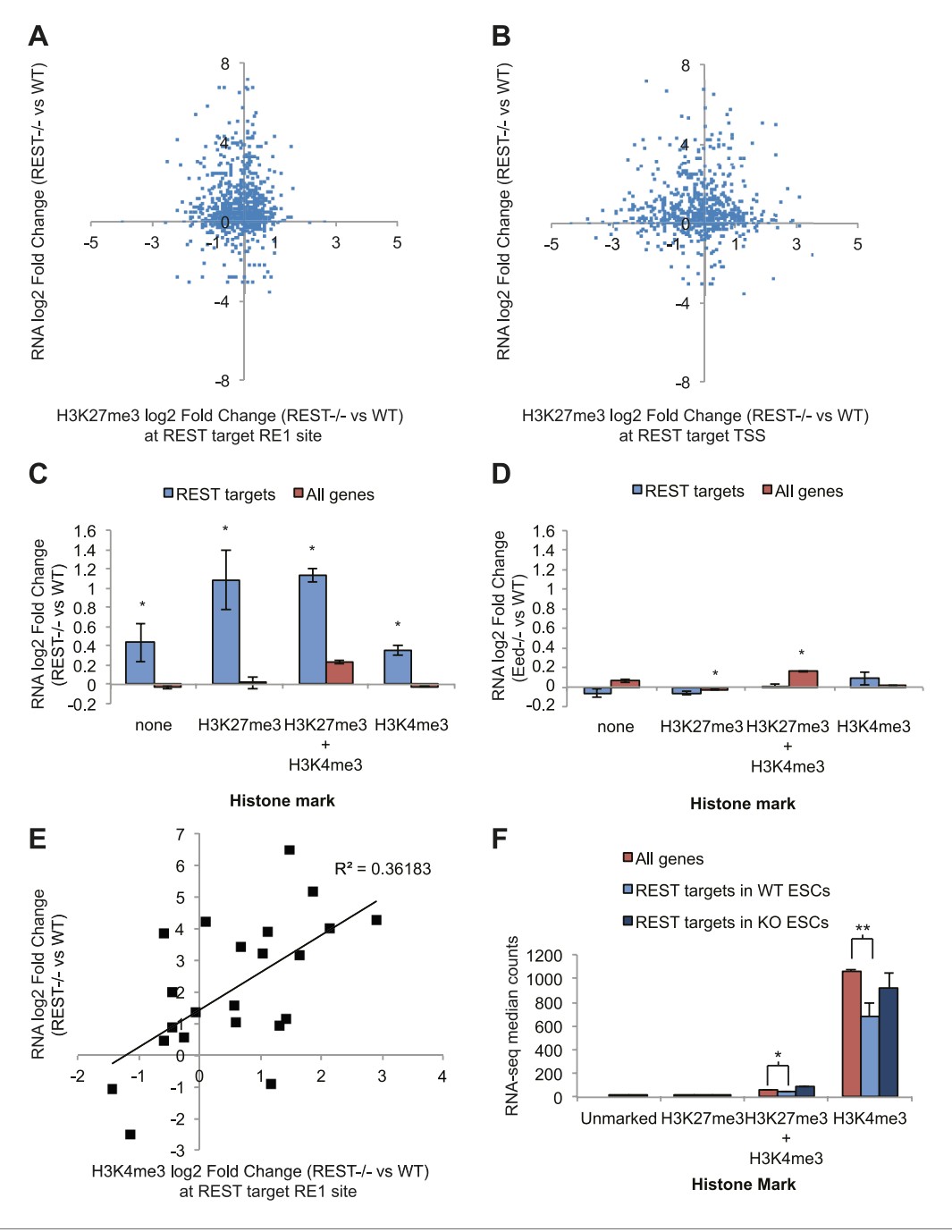

**Figure 3**. REST-dependent changes in expression of REST targets are correlated significantly with REST-dependent changes in H3K4me3, not H3K27me3. (**A**) RNA-seq log2(Fold Change) results for *Rest*[−/−] ESCs are not correlated with changes in H3K27me3 levels at REST sites or (**B**) REST target transcriptional start sites (TSS). (**C**) All REST target genes are de-repressed in *Rest*[−/−] ESCs regardless of H3K27me3 or H3K4me3 status. (**D**) In contrast, REST targets show no transcriptional changes in *Eed*[−/−] ESCs, which have highly reduced levels of H3K27me3, and genes with this mark show significant increases in expression ($p < 0.005$). (**E**) Changes in H3K4me3 enrichment in *Rest*[−/−] ESCs strongly correlate with REST target gene expression changes ($p < 0.01$). (**F**) Expression levels of H3K4me3-marked REST target genes are significantly reduced relative to H3K4me3-marked genes and de-repressed in *Rest*[−/−] ESCs (*$p < 0.05$, **$p < 0.001$).

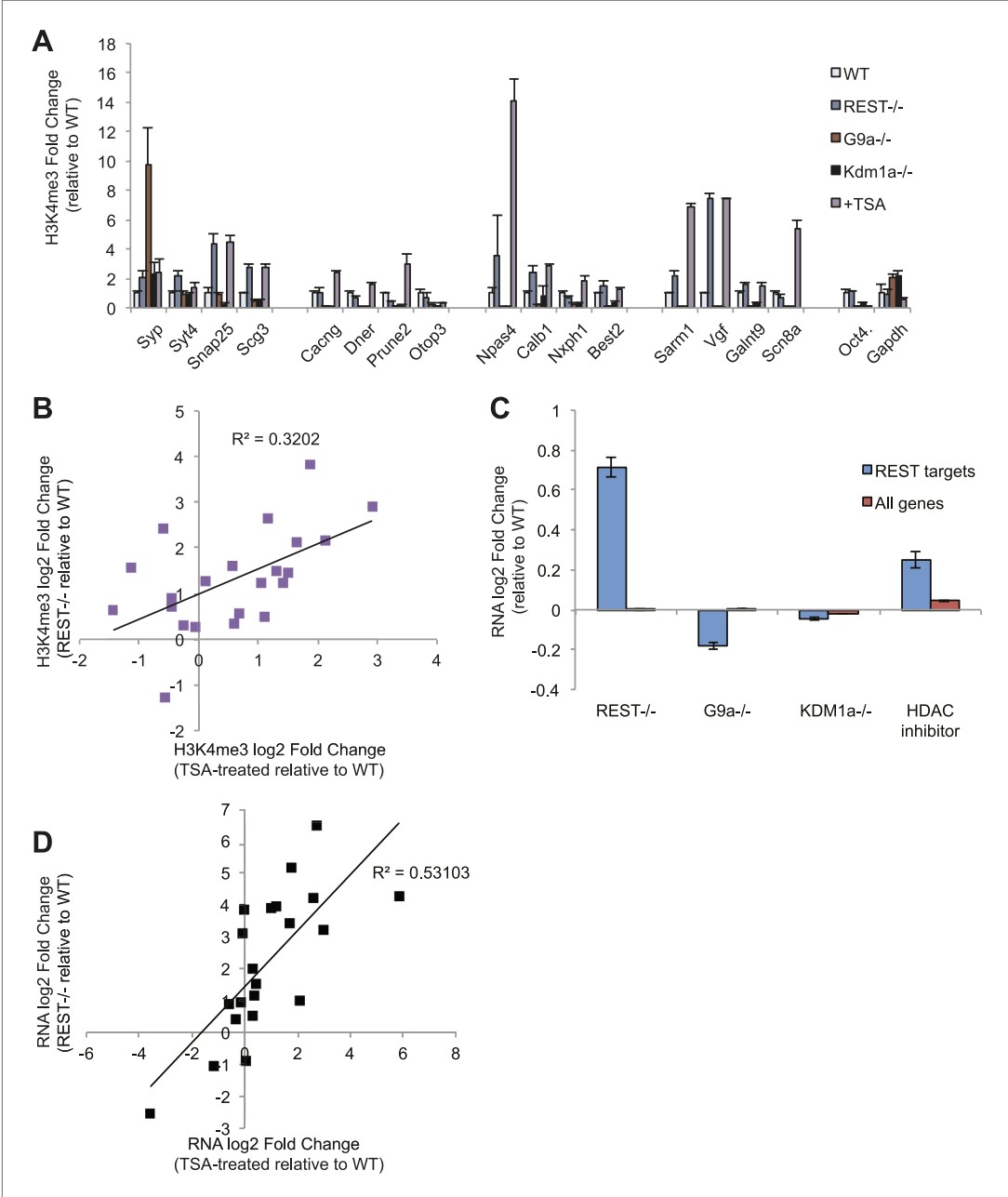

**Figure 4**. REST antagonizes H3K4me3 in ESCs through histone deacetylase activity. (**A**) Only treatment with the histone deacetylase inhibitor trichostatin-A (TSA) results in the increased H3K4me3 enrichment seen in *Rest⁻/⁻* ESCs. The active *Oct4* and *GAPDH* promoter regions that lack RE1 sites were included as control regions enriched for H3K4me3. (**B**) Changes in H3K4me3 enrichment at RE1 sites due to the loss of REST are significantly correlated with those due to TSA treatment (p < 0.01). (**C**) Microarray analysis reveals that HDAC inhibition by trichostatin-A (TSA) preferentially de-represses REST targets, unlike the loss of G9a or Kdm1a. (**D**) Changes in expression of select REST target genes due to REST loss significantly correlate with changes in expression due to HDAC inhibition with TSA (p < 0.001).

The following figure supplement is available for figure 4:

**Figure Supplement 1**. H3K9ac levels increase after TSA treatment.

(p < 0.01, *Figure 4B*), consistent with previous studies that have indicated a negative interaction between de-acetylation at lysine 9 by HDAC activity and trimethylation at lysine 4 (*Lee at al., 2006b*). As expected, we also observed elevated acetylated H3K9 levels at RE1 sites after TSA treatment

(*Figure 4—figure supplement 1*). Additionally, we utilized gene expression data sets published previously to analyze the transcriptional consequences of co-repressor removal. This analysis revealed that REST targets are de-repressed only in the absence of HDAC activity, but not when G9a and Kdm1a are mutated (*Figure 4C*). By focusing on those genes for which we observed H3K4me3 effects, we also found a significant correlation between the magnitude of the change in RNA levels when REST is deleted and that when histone deacetylase activity is strongly inhibited by TSA (p < 0.005, *Figure 4D*). These results suggest that REST repression in ESCs is mediated primarily by recruited HDACs that serve as a counterbalance to H3K4me3 levels and basal RNA polymerase II activity, although the nature of the cross-talk between HDACs and H3K4 trimethylation in this context awaits future investigation.

## Discussion

The Polycomb-mediated bivalent pattern of histone modifications, consisting of H3K4me3 and H3K27me3, has been proposed as a central mechanism for maintenance of a poised transcriptional status in undifferentiated stem cells. However, a study on early zebra fish embryos showed that only 36% of inactive gene promoters were associated with a bivalent histone modification pattern, while 28% exhibited enrichment of H3K4me3 alone, yet remained inactive (*Vastenhouw et al., 2010*). This suggests that in addition to PRC2, alternative repressor mechanisms exist for recruiting chromatin modifiers to poised genes, although such proteins have not been identified. We propose that the REST/HDAC repressor mechanism represents one such alternative mechanism for genes in the neuronal lineage. By extension, our results indicate that both neuronal cell fate determining genes and neuronal genes expressed later in the differentiation program are poised in ESCs, albeit by different mechanisms.

We identified three classes of REST-bound sites: 1) sites that lacked trimethylation at either H3K4 or H3K27, 2) sites that exhibited the bivalent modification pattern H3K4me3 and H3K27me3, and 3) sites marked by H3K4me3 only (*Figure 1—figure supplement 3D*). The first class, lacking trimethylation, was preferentially located distal to promoter regions. Preliminary studies on these distal sites show that some overlap with the enhancer mark H3K4me1 and/or H3K27ac (data not shown), potentially indicating a role for REST-directed repression at specified distal enhancer regions, an intriguing hypothesis that awaits future analysis. However, because it is currently unclear which promoters/genes these distal sites regulate, we focused on the other two subclasses, which were located within 20 kb of annotated TSSs. The majority of these REST-associated regions were enriched for H3K4me3, consistent with our idea that the balance of REST-recruited HDACs and H3K4me3 was sufficient to poise neuronal genes independent of Polycomb.

The REST-bound promoters with H3K27me3 raised the possibility of a functional link between REST and PRC2 in ESCs at these sites. Additionally, a previous report that the ncRNA HOTAIR can link REST and PRC2 suggested that ncRNA-mediated interactions in ESCs could result in PRC2 recruitment to REST-bound RE1 sites, and conversely, that REST complexes could be recruited to PRC2 bound regions independent from recognition of the RE1 motif (*Tsai et al., 2010*). Our results are not consistent with either of these scenarios. First, REST-binding sites within the ESC genome appeared to be dependent exclusively on the underlying DNA sequence, because REST binding was correlated strictly with the presence of RE1 motifs. Second, the Polycomb complex member Eed, which is required for H3K27me3 deposition (*Montgomery et al., 2005*), was absent from the REST complexes characterized by mass spectrometry and co-immunoprecipitation, undermining the likelihood of either repressive complex directly targeting the other. Third, only a minority of REST-bound sites was associated with H3K27me3 enrichment and PRC2 localization (~3%). Moreover, H3K27me3 was not preferentially enriched at regions with multiple RE1 sites and thus did not show strong association with REST. Finally, more than 97% of the RE1 sites associated with PRC2 in wild-type cells also showed H3K27me3 enrichment in *Rest*⁻/⁻ ESCs, including at gene promoters, indicating that REST was not required for PRC2 recruitment at these regions. Taken together, we conclude that REST and PRC2 act largely independently, even at shared target genes, in ESCs.

The term 'developmental regulators' has been used to describe Polycomb targets in ESCs (*Boyer et al., 2006*; *Lee et al., 2006a*). Therefore, we considered the possibility that the small subclass of REST/PRC2 targets might represent a specialized set of genes for promoting neural development. Gene ontology analysis, however, revealed no apparent distinction between biological functions in this subclass and the biological functions associated with the REST pattern alone. Both subclasses contained

genes known to influence neurodevelopment, many of which persist in the adult nervous system, as well as other categories considered to be late neuronal genes involved in mature neuronal function, such as synaptic components and voltage-gated channels. Thus, PRC2 does not specifically target regulators of neurodevelopment within the REST-regulated network of genes.

Different criteria used to define and quantify H3K27me3 domains may explain the discrepancies between our conclusions and those of others suggesting that REST mediates PRC2 recruitment in ESCs (*Dietrich et al., 2012*). A critical distinction is that our analysis defined H3K27me3-enriched regions before comparing the computed H3K27me3 signals between WT and *Rest*$^{-/-}$ ESCs. By applying this initial binary condition, our analysis avoids the contribution of fluctuations in background signal. We argue that comparing the computed H3K27me3 ChIP-seq signals at all REST sites without considering initial H3K27me3 background signal would always find some level of relationship between REST and PRC2 and therefore eliminate the null hypothesis a priori. Therefore, we limited our comparisons of H3K27me3 domains to regions that also showed the clear presence of PRC2 activity in WT ESCs. Additionally, the loss of REST may generate small but reproducible effects in measured histone modifications due to local changes in nucleosome density, rather than actual changes in a specific modification (*Zheng et al., 2009*). Similar to a recent study that showed the presence of REST evicts nucleosomes at RE1 DNA sequence motifs (*Valouev et al., 2011*), we evaluated in vivo nucleosome positioning in ESCs and found that phasing of nucleosomes centered at RE1 motifs was displaced in *Rest*$^{-/-}$ cells (data not shown). Therefore, although it can appear that PRC2 activity is increased specifically at RE1 sites by the loss of REST (*Dietrich et al., 2012*); this likely reflects a secondary consequence of the gain of a nucleosome at the RE1 site, due to the loss of the REST protein and subsequent fill-in of its footprint by a single histone octamer. How REST-associated nucleosome positioning generally affects gene expression is not yet known, but there are well-documented examples in lower eukaryotes of dynamic nucleosome positioning as a mechanism of gene regulation (*Bai and Morozov, 2010*).

In addition to maintaining nucleosome-depleted regions, our results indicate that REST likely counterbalances RNA Pol II activity primarily through recruitment of histone deacetylase activity in undifferentiated cells, echoing results observed in human T cells (*Zheng et al., 2009*). Acetylated histone tails have been shown to interact with bromodomains of transcription factors, such as Brd4, which promotes recruitment of Mediator complexes or positive transcription elongation factor b (P-TEFb) and release of paused RNA Pol II (*Jang et al., 2005*; *Yang et al., 2005*; *Wu and Chiang, 2007*). These interactions may explain the observed dependence of H3K4me3 on TSA and H3K9 acetylation.

The net transcriptional effect on genes in *Rest*$^{-/-}$ ESCs was variable and depended on the locus (*Johnson et al., 2008*; *Jorgensen et al., 2009*), which is likely due to specific activators being present or absent in ESCs, as well as additional repressive mechanisms that may be acting at the same target. However, changes in the H3K4me3 mark (and histone acetylation) due to the loss of REST were significantly correlated with changes in gene expression, while the other histone modifications we analyzed were not. This suggests that REST-directed repression of H3K4 methyltransferases or activation of H3K4me3 demethylases is important to restrict the amount of expression from these genes. A potential candidate demethylase is SMCX (Jarid1C), which binds REST in HeLa cells and can regulate promoter H3K4me3 levels (*Tahiliani et al., 2007*), although we found no evidence of SMCX binding in our mass spectrometry results. In addition, there is evidence that H3K4me3 'primes' non-expressed genes for acetylation and increased gene expression after histone deacetylase loss (*Wang et al., 2009*; *Lopez-Atalaya et al., 2013*). Thus, as neuronal differentiation proceeds and REST/HDAC levels on target chromatin decrease dramatically, those genes previously marked with H3K4me3 increase this mark simultaneously with H3K9 acetylation in a rapid feed-forward mechanism.

Based on our results, we propose that the loss of the REST or Polycomb repressor complexes from different sets of genes, in conjunction with the recruitment of transcriptional activators, allows for finely tuned, graded expression changes over the course of differentiation. In stem/progenitor cells, REST is a key repressor of genes crucial to the terminally differentiated neuron, while PRC2 is a repressor of a REST independent pathway regulating pro-neural genes that are required at earlier differentiation stages (*Mohn et al., 2008*). Finally, 'terminal selector' genes, which are transcriptional activators in mature neurons, also drive their own expression to maintain the terminally differentiated phenotype (*Hobert, 2011*). In a similar but reversed case, the *REST* gene, which itself contains a REST-binding site, may function to reduce its own expression so that differentiation can proceed unidirectionally.

It will be interesting in the future to see whether repressors in other cell lineages play similar roles in poising terminal genes in stem/progenitor cells. A recent study has suggested that structural genes

encoding mature cardiac cell functions are regulated primarily by transcriptional activators rather than by H3K27me3 (*Paige et al., 2012*). An alternative possibility, based on our study, is that these temporally delayed cardiac genes are repressed by factors, still to be identified, which recruit chromatin modifiers other than Ezh2 in order to balance the activation mark in stem cells. In neurons, direct reprogramming can occur by introducing pro-neural (Ascl1) along with terminal genes (e.g. Myt1l) into somatic cells (*Vierbuchen et al., 2010*) perhaps because they represent distinct regulatory pathways. Better knowledge of the factors regulating terminally differentiated gene chromatin could provide insight into the mechanisms underlying direct reprogramming of fibroblasts into different types of cells (*Nam et al., 2013*).

## Materials and methods

### Construction of Flag-BioT-mREST (pFBmR)

Mouse REST (mREST) CDS, lacking the start and stop codons and flanked by BamH1 sequences, was amplified from pcDNA3.1A(−)-mREST-Myc-His (Mandel, unpublished) using the following PCR primers: 5′- GCG CGG ATC CCC ACC CAG GTG ATG GGG CA -3′ (JL70112a) and 5′- GCG CGG ATC CCT ACT CCT GCT CCT CCC GC -3′ (JL70705a) (underlined are BamHI sites). The fragment was cloned into a TOPO-TA vector, released by BamH1, and then cloned in frame into the BamH I site in pEFrFLAG-BIOpGKpuropAv1 (pFL-Big) (*Wang et al., 2006*), kindly provided by Jianlong Wang and Stuart Orkin (Harvard Medical School).

### Generation of a Flag-BioT-REST-expressing ESC cell line

The pFL-Big plasmid was kindly provided by Jianlong Wang and Stuart Orkin (Harvard Medical School), the N6 and N8 ESC lines were provided by Zhou-Feng Chen, (Washington University in St. Louis). Plasmid pFBmR was linearized with Sca I and transfected with Lipofectamine 2000 (Invitrogen, Carlsbad, CA) into BirA-J1 ES cells that stably express the *Escherichia coli* Bir A ligase (*Wang et al., 2006*) (kindly provided by Jianlong Wang and Stuart Orkin) and maintained in 15% FBS in DMEM (#11965; Gibco) supplemented with penicillin/streptomycin, 2 mM L-glutamine, non-essential amino acids (#M7145; Sigma), 0.1 mM 2-mercaptoethanol, 8 mg/l adenosine, 8.5 mg/l guanosine, 7.3 mg/l uridine, 7.3 mg/l cytidine, 2.4 mg/l thymidine, and 10 U/ml LIF (# ESG1107; Chemicon/Millipore) on tissue culture plates coated with 0.1% gelatin (Sigma). Stable BioT-REST expressing cells were selected in 2 μg/ml puromycin and individual clones were hand picked under a microscope. The clones were then screened by Western blot analysis for REST protein level, using an antibody raised against the C-terminus of hREST (*Ballas et al., 2005*). One clone expressing REST at levels ~5 fold that of endogenous REST (clone #60) was used in the streptavidin pull-down experiment for mass spectrometric analysis.

### REST complex purification

ES cells expressing BioT-REST (clone #60) from 10, 15 cm dishes were used for each pull down. The cells were harvested and pelleted, then lysed in 2.5 ml cold lysis/binding buffer (0.5 mM EDTA, 150 mM NaCl, 0.5% Triton X-100, 10% glycerol, 1 mM NaF, 1 mM $Na_3VO_4$, 0.5 mM DTT in pH 7.5, 50 mM Tris–Cl with 1× Roche complete protease inhibitors cocktail) with the help of sonication on ice (4 rounds of 20 strokes, output 4, 40% duty cycle, Sonifier). The cell lysate was cleared by centrifuge at 4°C and incubated with buffer-exchanged 200 μl streptavidin M-280 magnetic beads (Dynal beads/Invitrogen) at 4°C for 3 hr in lysis/binding buffer. After the incubation, the beads were pelleted and washed three times with 1 ml cold lysis/binding buffer and three times with 1 ml cold PBS. The beads were then eluted twice with 200 μl and 100 μl elution buffer (1:1 (vol/vol) acetonitrile/$H_2O$ in 0.1% trifluoroacetic acid) at 65°C for 10 min. The two eluates were combined and SpeedVac dried under no heat and subjected to MudPIT analysis as powder. The parental BirA-J1 ES cells, which express *E. coli* BirA ligase but no BioT-tagged REST, were processed and analyzed in parallel as the negative control pull down.

### Proteomic analysis of REST complex

The eluted REST complex was solubilized in 8 M urea containing 10 mM dithiothreitol and incubated at 60°C for 30 min. The solution was cooled to room temperature and iodoacetamide was added to a final concentration of 15 mM and incubated at room temperature for 20 min in dark. The solution was then diluted to a final urea concentration of 2 M with 100 mM Tris–HCl. The proteins were digested with 1 μg of trypsin at 37°C overnight. The digestion was terminated by adding formic acid to 5%, and centrifuged. Half of the peptides containing supernatant were used for liquid chromatography coupled with mass spectrometry analysis to identify proteins. Peptides from each pull-down sample were

pressure-loaded onto a 250 μm i.d. fused silica capillary column packed with a 3 cm, 5 μm Partisphere strong cation exchanger (SCX, Whatman, Clifton, NJ) and a 3 cm, 10 μm Aqua reversed-phase C18 material (Phenomenex, Ventura, CA), with the SCX end fritted with immobilized Kasil 1624 (PQ Corperation, Valley forge, PA). After desalting, a 100 μm i.d. capillary with a 5 μm pulled tip packed with a 10 cm, 54 μm Aqua C18 material was attached to a ZDV union, and the entire split-column was placed inline with an Agilent 1100 quaternary HPLC (Agilent, Palo Alto, CA) and analyzed using a modified, six-step multi-dimensional protein identification technology (MudPIT) described previously (*Washburn et al., 2001*). As the peptides were eluted from the microcapillary column, they were electrosprayed directly into an LTQ linear ion trap mass spectrometer (ThermoFinnigan, San Jose, CA) with the application of a distal 2.5 kV spray voltage. A cycle of one full-scan mass spectrum (400–1400 m/z) followed by 5 data dependent MS/MS scan at a 35% normalized collision energy was repeated continuously throughout each step of the multidimensional separation. The resulting MS/MS spectra were searched with the SEQUEST algorithm (*Griffin et al., 1995*) against a mouse IPI database (version 3.30, released at 28 June 2007) that was concatenated to a decoy database in which the sequence for each entry in the original database was reversed. The search parameters include a static cysteine modification of 57 amu and no trypsin specificity. The database search results were assembled and filtered using the DTASelect program (*Tabb et al., 2002*) requiring a protein level false discovery rate less than 1%, all peptides identified are required to be tryptic, and at least two peptides are required for a protein to be identified. Under such filtering conditions, no peptide hit from the reverse database was found.

## Cell culture

Mouse ESCs cultures, N6 (WT) and N8 (*Rest*$^{-/-}$) (*Jorgensen et al., 2009*), were cultured in DMEM medium described above. ESCs were cultured on feeder layers of irradiated mouse embryonic fibroblasts and passaged three times on plates coated with 0.1% gelatin to eliminate MEFs before harvesting cells for RNA or chromatin purification.

## RNA-seq sample preparation and analysis

Total RNA was extracted using TRIzol (Invitrogen) followed by on-column DNAse treatment with RNase-free DNase and RNesay mini kit (Qiagen). 2 μg total RNA was used to make one sequencing library. Two biological replicates were made for each condition: WT ESC and *REST*$^{-/-}$ ESC. Indexed libraries were prepared using the Illumina TruSeq RNA Sample Preparation Kit v2 (San Diego, CA). Four libraries were mixed at equal concentration and sequenced by an Illumina HiSeq 2000 sequencer at the OHSU Massively Parallel Sequencing Shared Resource (MPSSR). Reads were mapped using Subread (*Liao et al., 2013*) and gene counts assigned using FeatureCounts (*Liao et al., 2014*). Differential expression analysis was performed using edgeR (*Robinson et al., 2010*) with p-values assessed by both tag-wise and common dispersion analysis. Primary reads and mapped gene counts can be found at GSE59442.

## mRNA expression analysis

Total RNA was isolated by disrupting cultured cells with Trizol reagent (Invitrogen) followed by chloroform extraction and ethanol precipitation/wash according to the manufacturer's instructions. For each sample, 1 μg of purified RNA was used as template for first strand cDNA synthesis with random hexamer primers and SuperScript III reverse transcriptase (Invitrogen) following the standard manufacturer's protocol. cDNA quantities were evaluated by quantitative real-time PCR measuring SYBR Green fluorescence on an ABI 7900HT. Following activation of the hot start polymerase at 95°C for 10 min, reactions were cycled 40 times at 95°C for 15 s and 60°C for 1 min. Experimental cDNA samples were run in triplicate. Primer sequences used for amplification are listed in *Table 1—Source data 1*. Relative gene expression for genes of interest (GOI) was calculated using the ΔΔCt method and normalized to Gapdh levels to control for variation in reaction inputs. Standard deviation of the normalized expression was calculated as; $SD = (\text{normalized value}) \times \ln(2) \times \sqrt{(SD_{Gapdh})^2 + (SD_{GOI})^2}$.

## Chromatin Immunoprecipitation (ChIP) analysis

ChIP analyses were performed as described previously (*Ballas et al., 2005*). Briefly, cells were treated with 1% formaldehyde for 10 min at RT to form protein–DNA crosslinks. Crosslinking reaction was quenched by addition of glycine to a final concentration of 0.125 M and incubating for 5 min at RT, followed by two washes with PBS. Harvested cells were resuspended in nuclei isolation buffer (5 mM HEPES, pH 8.0, 85 mM KCl, and 0.5% Triton X-100) and incubated for 10 min on ice. Pelleted nuclei were resuspended in

nuclei lysis buffer (50 mM Tris–HCl, pH 8.0, 10 mM EDTA, and 1% SDS) at an approximate concentration of $10^7$ cells per ml prior to shearing chromatin by sonication to a final size range of ~100–750 bp. Chromatin lysate was diluted 1:10 with ChIP dilution buffer and specific antibodies were added for overnight incubation at 4°C. The following antibodies were used for immunoprecipitations: anti-H3K4me3 (07-473; Millipore), anti-Ac-H3K9 (H9286; Sigma), anti-2Me-H3K9 (ab1220; Abcam), anti-H3K27me3 (9733; Cell Signaling), anti-H3 (2650; Cell Signaling), anti-REST-C (*Ballas et al., 2005*; *Otto et al., 2007*), anti-Ezh2 (5246; Cell Signaling), LSD1 (Kdm1a) antibody from Yang Shi (Harvard Medical School). Protein A conjugated magnetic beads that had been blocked with BSA were used to purify immunocomplexed chromatin fragments by incubating with sample lysates for 3 hr at 4°C. Beads were sequentially resuspended in low salt, high salt, and LiCl wash buffers followed by two final washes in TE buffer. Immunoprecipitated chromatin was eluted from the beads resuspended in elution buffer (50 mM Tris HCl, pH 8.0, 100 mM $NaHCO_3$, 1% SDS, and 200 mM NaCl) during reversal of formaldehyde crosslinks by overnight incubation at 65°C. Elutions were treated with RNase A (1 hr at 37°C) and proteinase K (2 hr at 55°C) prior to a final purification of DNA by column chromatography (Qiagen PCR Purification). Quantities of immunoprecipitated DNA were measured relative to signal from input samples by real-time PCR and analyzed using the $\Delta\Delta$Ct method. Primer sequences used for ChIP analysis are listed in *Table 1—source data 1*.

## Statistical analysis

Data were analyzed using linear regression analysis (*Figures 3E*, *Figure 4B,D*), or Student's *t*-test (all other figures). A threshold of $p < 0.05$ was interpreted as significant.

## ChIP Seq analysis

ChIP-isolated DNA was pooled (three technical replicates done in parallel from each of the two independent biological replicates) and fragments were processed to blunt ends followed by A-tailing to facilitate ligation of Illumina oligo adapters. PCR amplification was run for 12–14 cycles with primers complementary to adapter sequence to amplify the pool of ChIP DNA with addition of the adapter sequence. PCR products in the range of 200–300 bp were isolated by agarose gel electrophoresis followed by gel extraction. 5 ng of sheared DNA purified from chromatin samples without immunoprecipitation was also processed in this manner as an input control. DNA fragments were sequenced using the Illumina Genome Analyzer II platform. The number of unique reads aligned to the mm9 assembly for each ChIP-Seq was: REST (WT ESC) 11,954,736, H3K27me3 (WT ESC) 17,893,323, H3K27me3 (*Rest*$^{-/-}$ ESC) 14,102,052, and Input (WT ESC) 12,092,824. Raw reads were aligned with Bowtie and only uniquely mapped reads were kept. After alignment, PeakRanger (*Feng et al., 2011*) and MACS (*Zhang et al., 2008*) were used to call peaks and the overlap peak set was retained. Overlapped regions may have different boundaries. To identify H3K27me3 peaks conserved between cell lines and called by independent means, we used only those MACS-called peaks that overlapped between our H3K27me3 peak set and those identified by the Encode project (*Grant et al., 2011*). To find the genomic regions with increased levels of H3K27me3 after REST knock out, the H3K27me3 ChIP in REST knock-out is used as the 'treatment' and H3K27me3 ChIP in wild-type as the 'control' for both of the two software programs and the same significance threshold was set for both. The data sets were then swapped and analyzed for regions with decreased levels of H3K27me3. To get the reads for histograms shown in *Figure 1—figure supplement 3A*, the 'wig' module of PeakRanger parsed all aligned reads and counted reads within the specified regions. To find the REST-binding motif (RE1), REST peak coordinates were used as input for FIMO (*Grant et al., 2011*). Previously published ChIP-seq data sets used in the analysis of H3K27me3 were GSE51006 (*Ferrari et al., 2014*), GSE48172 (*Hu et al., 2013*), GSE27341 (*Arnold et al., 2013*), and GSE49431 (*Kaneko et al., 2013*). Previously published ChIP-seq data set used in the analysis of PRC2 components was GSE49431 (*Kaneko et al., 2013*). Previously published ChIP-seq data sets used in the analysis of REST complex components were GSE27841 (*Whyte et al., 2012*) and GSE24841 (*Williams et al., 2011*). Previously published ChIP-seq data set for H3K4me3 peaks was GSM1003756 (*Stamatoyannopoulos et al., 2012*). Data were collected as or converted to bigwig format using a combination of BEDtools (*Quinlan and Hall, 2010*), SAMtools (*Li et al., 2009*), and analyzed using the bigWigAverageOverBed module from the Kent source tools available from the UCSC genome browser (*Kent et al., 2010*).

## Gene ontology analysis

Gene lists derived from methods above and previous publications (*Young et al., 2011*) were formatted and uploaded to the AMIGO GO Enrichment tool and analyzed for enrichment in biological processes (*Carbon et al., 2009*).

Funding was provided by the Howard Hughes Medical Institute and the National Institutes of Health.

# Additional information

## Funding

| Funder | Grant reference number | Author |
|---|---|---|
| Howard Hughes Medical Institute | Gail Mandel: NS22518 | James C McGann, Jon A Oyer, Saurabh Garg, Huilan Yao, Jun Liu, Gail Mandel |
| National Institutes of Health | James C McGann: NS078886 | James C McGann |

The funders had no role in study design, data collection and interpretation, or the decision to submit the work for publication.

## Author contributions

JCMG, JAO, Conception and design, Acquisition of data, Analysis and interpretation of data, Drafting or revising the article; SG, LL, Acquisition of data, Analysis and interpretation of data; HY, Acquisition of data; JL, Conception and design, Acquisition of data, Analysis and interpretation of data; XF, JRY, Analysis and interpretation of data; GM, Conception and design, Analysis and interpretation of data, Drafting or revising the article

# Additional files

## Major datasets

The following datasets were generated:

| Author(s) | Year | Dataset title | Dataset ID and/or URL | Database, license, and accessibility information |
|---|---|---|---|---|
| Oyer JA, McGann JC, Feng X, Mandel G | 2014 | Polycomb Repressor Complex 2 and REST-associated histone deacetylases are independent pathways toward a mature neuronal phenotype | GSE48320 | Publicly available at the NCBI Gene Expression Omnibus (http://www.ncbi.nlm.nih.gov/geo/). This work is licensed under the Creative Commons Attribution 3.0 Unported License. |
| McGann JC, Mandel G | 2014 | Polycomb Repressor Complex 2 and REST-associated histone deacetylases are independent pathways toward a mature neuronal phenotype | GSE59442 | Publicly available at the NCBI Gene Expression Omnibus (http://www.ncbi.nlm.nih.gov/geo/). This work is licensed under the Creative Commons Attribution 3.0 Unported License. |

**Reporting standards:** Standard used to collect data: NCBI Geo Accesion database standards for ChIP-seq and RNA-seq data were followed.

The following previously published datasets were used:

| Author(s) | Year | Dataset title | Dataset ID and/or URL | Database, license, and accessibility information |
|---|---|---|---|---|
| Ferrari K, Scelfo A, Jammula SG, Pasini D | 2013 | Polycomb-dependent H3K27me1 and H3K27me2 regulate active transcription and enhancer fidelity | GSE51006 | http://www.ncbi.nlm.nih.gov/geo/. |

| | | | | | |
|---|---|---|---|---|---|
| Gao X, Hu D, Shilatifard A | 2013 | Mll2 branch of the COMPASS family regulates bivalent promoters in mouse embryonic stem cells | GSE48172 | | http://www.ncbi.nlm.nih.gov/geo/. |
| Schoeler A, Stadler M, van Nimwegen E, Schübeler D, Beisel C | 2012 | Role of REST during neuronal differentiation | GSE27341 | | http://www.ncbi.nlm.nih.gov/geo/. |
| Kaneko S, Bonasio R, Reinberg D | 2013 | ChIP-seq for PRC2 components and H3K27me3 in E14 mouse embryonic stem cells | GSE49431 | | http://www.ncbi.nlm.nih.gov/geo/. |
| Whyte W, Bilodeau S, Hoke H, Orlando DA, Frampton GM, Young RA | 2012 | Enhancer Decommissioning by LSD1 During Embryonic Stem Cell Differentiation (ChIP-seq) | GSE27841 | | http://www.ncbi.nlm.nih.gov/geo/. |
| Williams K, Pedersen MT | 2011 | Tet1 and hydroxymethylcytosine in transcription and DNA methylation fidelity (ChIP/DIP-Seq data) | GSE24841 | | http://www.ncbi.nlm.nih.gov/geo/. |
| Yue F, Cheng Y, Breschi A, Vierstra J, Wu W, Ryba T, Sandstrom R, Ma Z, Davis C, Pope BD, Shen Y, Pervouchine DD, Djebali S, Thurman B, Kaul R, Rynes E, Kirilusha A, Marinov GK, Williams BA, Trout D, Amrhein H, Fisher-Aylor K, Antoshechkin I, See L, Fastuca M, Drenkow J, Zaleski C, Dobin A, Prieto P, Lagarde J, Bussotti G, Tanzer A, Denas O, Li K, Bender MA, Zhang M, Byron R, Groudine MT, McCleary D, Pham L, Ye Z, Kuan S, Edsall L, Wu Y, Rasmussen MD, Bansal MS, Keller CA, Morrissey CS, Mishra T | 2012 | Stanford_ChipSeq_ES-E14_H3K4me3_std | GSM1003756 | | http://www.ncbi.nlm.nih.gov/geo/. |

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
