## [Decision Letter]

Thank you for sending your work entitled “Polycomb and REST-associated histone
deacetylases are independent pathways toward a mature neuronal phenotype” for
consideration at *eLife.* Your article has been evaluated by a Senior
editor and 3 reviewers, one of whom is a member of our Board of Reviewing Editors, and
one of whom, Ramin Shiekhattar, has agreed to reveal his identity.

The Reviewing editor and the other reviewers discussed their comments before we reached
this decision, and the Senior editor has assembled the following comments to help you
prepare a new submission.

Although one of the reviewers was positive, two other reviewers raised substantial
concerns that preclude acceptance at this time, but we encourage you to read the
reviewers’ comments and submit a new manuscript (full comments below). The main issues
are related to the biochemistry, where we feel you can easily solve this problem by
performing some more experiments as stated by reviewer #1. Reviewer #2 felt that some of
the conclusions were not supported by the presented data. Having said this, we believe
that a revised manuscript addressing the comments of reviewers 1 & 2 will result in
an important addition to the field. The previous data suggesting that Rest and PRC2
physically interact are not conclusive and settling the issue is important, and so we
look forward to receive a new manuscript addressing the issues raised.

Reviewer #1:

The manuscript by Oyer et al. reports the observation that different chromatin
regulators drive pro-neural and terminal differentiation of neuronal lineages. The
authors push the idea that REST and Polycomb are temporally regulated to push the
differentiation of ES cells towards a gradually more neuronal fate. In this manuscript
the authors describe their attempt to see if there is a link between PRC2 and REST and
test the idea of whether these two repressive complexes are working synergistically or
in tandem to maintain a more ES cell-like fate and prevent the expression of neuronal
genes. The authors used a variety of approaches to identify ESC-specific factors
associated with REST, the relationship between H3K27me3 and REST occupancy at target
genes and the biological relevance of the loss of REST during differentiation.

The observations presented in this manuscript are interesting, however, there are some
points that need clarification and additional experiments to strengthen the claims put
forth in the paper. Publication should be pending on whether these points are addressed.
The comments of the paper are listed below:

1) During the purification of the REST-associated complexes, the authors do a nice job
of validating the candidates by ChIP-qPCR. However, the peptide counts for all of the
targets including the bait were relatively low and this may preclude the presence of
other interacting protein(s) that may be of interest. More rigorous biochemical
purification would benefit the authors in their search for the missing deacetylase,
therefore either increasing the number of ESC plates used in the purification should
improve the identification of novel factors or the authors should try a different tandem
affinity purification strategy to improve the identification of interacting
proteins.

2) The use of TSA by the authors as a proxy of histone deacetylation activity should be
supported with a figure showing at least the effect of knocking down HDAC1/2, which was
identified in from their proteomic analysis.

3) In the Discussion, the authors make a statement regarding the link between REST and
PRC2. The statement that “Polycomb family proteins were not present in the REST
complexes characterized by mass....” This is not valid reasoning as to why REST and PRC2
are acting independently. These two complexes do not need to be associating together in
order to function at same targets.

4) During the purification of the REST complexes, the authors used sonication. This
could be a reason why the REST and PRC2 link was not detected by the authors. The
association of these two complexes may be nucleic acid dependent and sonication may be
disrupting this interaction. The authors need to repeat their purifications using
another method to disrupt the cells and cell fractionated to enrich for the nuclear
fractions, not whole cell lysates.

Reviewer #2:

In this manuscript, Oyer and collaborators investigated the role of the repressor
complex REST in silencing neuronal specific genes in embryonic stem cells (ESCs).
Silencing of lineage specific genes in ESCs is often performed by the Polycomb group of
proteins. Yet the promoters of those genes are frequently decorated with
Polycomb-specific marks (H3K27me3) as well as with active marks (H3K4me3) and are
referred to as “bivalent promoters”. Previous works have highlighted the synergistic
actions of Polycomb and REST in different cellular model systems (including ESCs; see
Science 329, 689, 2010; MCB 31, 2100, 2011; PLOS Gen 8, e1002494, 2011). Oyer and
collaborators now report here that the vast majority of REST target genes are not
decorated with H3K27me3, and that MS analyses have suggested that Polycomb proteins and
REST are not part of the same complex, thus challenging the notion that Polycomb and
REST act together in the regulation of neuronal specific genes. They favor a model in
which REST acts in combination with HDAC, G9A and KDM1a. Although this topic is of great
interest, I find that many of the authors' conclusions are not well supported by the
experimental data presented. More stringent and appropriate controls, and more
experimental settings, may help to solidify their conclusions.

1) Very surprisingly, the authors do not comment on why there is a dramatic increase (of
almost 40%!) of H3K27me3 target genes in Rest-/- cells as compared to wild-type ESCs.
Additionally, the bioinformatic analyses of REST ChIP-seq are quite poor overall.

2) It was previously published that REST and Polycomb are co-recruited via the ncRNA
HOTAIR. It is thus possible that the PcG and REST do not necessary occupy the same
nucleosome. All the analysis performed in Figure 1 is based on a “peak” overlap rather than looking at regions or genes.

3) The authors should also analyze PRC1, since CBX proteins have been demonstrated to
interact with REST.

4) The data presented in Table 2 are very
confusing. The H3K4me3 ChIP-seq is not mentioned and not properly analyzed in this
manuscript, yet it is part of this table. It would be interesting to overlap REST
ChIP-seq with H3K4me3 ChIP-seq and include this data to Figure 1.

5) Data presented in Figure 1 should be
normalized for nucleosome density (such as for histone H3).

6) The conclusions from Figure 2 are vague and
not supported by the data presented. The authors state that “the presence or absence of
the 3Me-H3K4 mark is an important part of the chromatin signature orchestrated by REST”.
Deletion of Rest affected the H3K4me3 mark of each set of genes in all possible
ways.

7) Similarly, for data presented in Figure 3,
Rest deletion affected gene expression of ESCs, EBs and neuronal precursors in all
possible ways. I couldn't find a common trend in any set of genes. The authors do not
describe properly the results obtained or the model system they used. For a reader not
familiar with the ESC differentiation protocols, it would be impossible to understand
the rationale behind these experiments.

In sum, there is no difference between the 4 groups of genes analysed in Figure 3. If the authors' only conclusion is that
deletion of the Rest repressor leads to an increase H3K4me3, this is in my opinion
already well-demonstrated. In any case, the author should comment on the fact that there
are no differences in expression of the genes analysed in Figure 3 between wild-type and Rest-/- mature neurons, which is
quite surprising.

It is very far fetched to draw any general conclusion when all the analyses performed in
Figure 3 are based on 12 genes. And it is
unclear why the authors used only 12 of the 14 genes presented in Figure 3.

8) In my opinion the only interesting (novel) data in this manuscript are those
presented in Figure 4: the authors should start
from here and develop this story further!

For example, they should perform this analysis in a genome-wide manner, with overlapping
RNA-seq analysis in wild-type and REST -/- cells in the 4 stages (including also mature
neurons).

9) Once again, the conclusions of Figure 5 are vague and not supported by the authors'
data: “These results suggest that REST repression in ESCs near a TSS is mediated
primarily by recruited HDACs that serve as a counterbalance to basal RNA polymerase II
activity, and that there is cross talk between HDACs and H3K4 trimethylation”.

The effect of TSA could be completely independent of REST. The correlation with H3K4me3
has nothing to do with REST. This figure only shows correlation between H3K4me3 and
histone acetylation.

10) The model presented in Figure 5D is wrong. Sox2 is not “bivalent” in ESCs, it is
actually expressed. While is not expressed in neurons.

---

## [Author Response]

We submitted previously to *eLife* a manuscript testing the roles of REST
and Polycomb repression in regulating the neuronal lineage in mouse embryonic stem
cells. This is an important question because Polycomb has been reported as a general
modulator of lineage control in pluripotent cells. As such, many studies focus just on
this complex to understand how genes are globally repressed in pluripotency. We provided
evidence indicating that the repressor REST is equally efficacious in this role and
likely works independently and in parallel for genes expressed in the neuronal lineage.
Two of the reviewers indicated that more work was required to substantiate our
conclusions, and the editors requested submission of a new manuscript addressing these
concerns. We took time to repeat a mass spectrometry study under different conditions,
performed new RNA-seq analysis, re-did some of the bioinformatics, and added other new
experiments as well. Consequently, this is a much more in-depth and solidified study
than the original work and still contains, we feel, the novelty component expected for
*eLife*.

Reviewer #1:

*1) During the purification of the REST-associated complexes, the authors do a
nice job of validating the candidates by ChIP-qPCR. However, the peptide counts for
all of the targets including the bait were relatively low and this may preclude the
presence of other interacting protein(s) that may be of interest. More rigorous
biochemical purification would benefit the authors in their search for the missing
deacetylase, therefore either increasing the number of ESC plates used in the
purification should improve the identification of novel factors or the authors should
try a different tandem affinity purification strategy to improve the identification
of interacting proteins*.

Our peptide counts were not dissimilar to other examples in the current literature where
novel interacting proteins were identified (Ding, et al, 2012, Cell Res; Costa et al,
2013, Nature), and we identified all known REST interactors using our approach.

However, in response to this concern, we increased the number of plates as suggested and
in addition followed the advice in point 4 (below) and performed mass spectrometry
analyses using nuclear extracts produced by dounce homogenization, rather than sonicated
whole cell extracts as before. Unfortunately, in two biological replicates of the
nuclear extract mass spec analysis, we identified primarily nucleolar proteins and no
peptides of known REST cofactors, including Rcor1, LSD1/Kdm1a, or G9a, which have been
published by several labs as an integral part of a bona fide REST complex and which we
identified readily with our previous method. Indeed, to our knowledge no one has
identified REST interactors with a REST pull-down using the Dignam method. Therefore, to
further test our negative result from the MudPiT analysis we performed a candidate
co-immunoprecipitation analysis using nuclear extracts that was not included in the
previous submission to confirm these potential interactions. The co-immunoprecipitation
analysis confirmed binding of the known interactors of REST but it also identified
members of the PRC2 complex Suz12 and Jarid2. However, neither Eed nor Ezh2 were present
in the REST pull-downs with this method (new Figure 1—figure supplement 2). The lack of Eed and Ezh2 is entirely consistent with
the lack of REST binding sites marked with H3K27me3, as they are both required for
deposition of the H3K27me3 mark. We have included the new results and a discussion of
the findings. Finally, we are not sure which deacetylase the reviewer refers to as
missing? Perhaps he meant the H3K4me3 demethylase? If the latter, it did not turn up in
either of our mass spec analysis methods, indicating that this may be an indirect effect
of REST-directed HDAC activity.

*2) The use of TSA by the authors as a proxy of histone deacetylation activity
should be supported with a figure showing at least the effect of knocking down
HDAC1/2, which was identified in from their proteomic analysis*.

We have provided a new figure (Figure 4—figure supplement 1) that shows increased levels of H3K9 acetylation at REST binding
sites after TSA treatment, which directly inhibits the enzymatic activity of HDACs. We
preferred this over the HDAC1/2 knock down experiment because knock down of HDACs is
likely to be partial, and we are not sure how we would interpret a negative result. In
addition, HDAC1/2 double knockout ESCs shows a profound loss of cell viability (Dovey et
al, 2014, PNAS). However, after deletion, these cells show increases in H3K9 acetylation
that are similar to those observed in our TSA treated cells. In addition, this report
also shows de-repression of several canonical REST targets including neuronal β tubulin,
synapsin, and VGF, consistent with our results. We include this new information in the
Results section.

With this, we hope the reviewer will give us a pass for the HDAC knock down
experiment.

*3) In the Discussion, the authors make a statement regarding the link between
REST and PRC2. The statement that “Polycomb family proteins were not present in the
REST complexes characterized by mass....” This is not valid reasoning as to why REST
and PRC2 are acting independently. These two complexes do not need to be associating
together in order to function at same targets*.

We apologize for our wording. Previous groups had suggested a direct role for REST in
recruiting PRC2 to its targets, as well as the possibility that PRC2 directly recruited
REST to its targets. Both of these scenarios required biochemical interaction. To better
clarify our meaning, we have replaced the sentence in question with the following:
“Second, the Polycomb complex member Eed, which is required for H3K27me3 deposition, was
absent from the REST complexes characterized by mass spectrometry and
co-immunoprecipitation, undermining the likelihood of either repressive complex directly
targeting the other.”

*4) During the purification of the REST complexes, the authors used sonication.
This could be a reason why the REST and PRC2 link was not detected by the authors.
The association of these two complexes may be nucleic acid dependent and sonication
may be disrupting this interaction. The authors need to repeat their purifications
using another method to disrupt the cells and cell fractionated to enrich for the
nuclear fractions, not whole cell lysates*.

Please see response to point 1, above.

Reviewer #2:

*1) Very surprisingly, the authors do not comment on why there is a dramatic
increase (of almost 40%!) of H3K27me3 target genes in Rest-/- cells as compared to
wild-type ESCs. Additionally, the bioinformatic analyses of REST ChIP-seq are quite
poor overall*.

Thank you for bringing this to our attention. Addressing this issue further solidified
our findings. To address the problem we did the following: First, we went back and
completely reanalyzed our results in a manner independent of using Peak Ranger, using
MACS only. Secondly, we re-analyzed our H3K27me3 peaks with consideration for how they
overlapped with the dataset provided by the ENCODE project’s H3K27me3 ChIP-seq in mESCs,
from Bing Ren at LICR. Only those peaks that were called in both our WT dataset and the
LICR dataset were deemed ‘true’ H3K27me3 peaks. To ensure that this was an accurate
representation of the peaks, we compared the read depth at these sites with four
additional published datasets and were satisfied that they were highly correlated and
were therefore valid (Figure 1—figure supplement 4). Importantly, with this new analysis, there is no difference in H3K27me3
peaks between WT and REST-/-cells. This result is now entirely consistent with our
approach showing lack of correspondence between REST binding sites and PRC2 activity. We
have modified the text to state the results and how we did the analysis more
clearly.

*2) It was previously published that REST and Polycomb are co-recruited via the
ncRNA HOTAIR. It is thus possible that the PcG and REST do not necessary occupy the
same nucleosome. All the analysis performed in*
Figure 1
*is based on a “peak” overlap rather than looking at regions or
genes*.

In response to this concern, we extended the REST ‘peak’ 1kb up- and downstream (Broad
REST sites) and re-ran the analysis. While the number of overlaps between PRC2 and REST
does increase (57 to 270), this quantity is still a very small minority of H3K27me3
sites (2.3%) and REST sites (12.6%). The additional analysis is shown in Figure 1 and described in the text.

*3) The authors should also analyze PRC1, since CBX proteins have been
demonstrated to interact with REST*.

In response to reviewer 1 we re-did the proteomics using a different approach and there
was still no evidence for PRC1 components in either analysis. We were aware of the CBX
result (Ren and Kerppola, 2011, Mol. Cell. Biol.), but it is not interpreted easily
because the effects of REST knockdown were opposite depending upon the chromatin
context. Further, our current study is focused on PRC2 components and their relationship
to REST regulation, a topic of current interest with respect to whether PRC2 alone
mediates bivalent chromatin required for pluripotency of all lineages.

*4) The data presented in*
Table 2
*are very confusing. The H3K4me3 ChIP-seq is not mentioned and not properly
analyzed in this manuscript, yet it is part of this table. It would be interesting to
overlap REST ChIP-seq with H3K4me3 ChIP-seq and include this data to*
Figure 1.

Table 2 is no longer included in the
manuscript. This is because we took the revision as an opportunity to update the
definitions of H3K4me3 and H3K27me3-marked promoters from Mikkelsen et al (2007, Nature)
to Young et al. (2011, Nucleic Acids Res.). After including the new definitions, there
is no gene ontology difference within REST targets based upon these marks. Regarding the
comment about H3K4me3 ChIP-seq, we have overlapped H3K4me3 peaks ([5], Genome Biology) with our REST ChIP
seq results and found that 37% of the REST sites located within 20kb of a TSS were also
marked by H3K4me3, and 62% of those within 5kb were marked by H3K4me3. We have included
this new information and it strengthens our idea that REST targets in ESCs are
associated with H3K4me3, as this reviewer already acknowledges in point 7 below.

*5) Data presented in*
Figure 1
*should be normalized for nucleosome density (such as for histone
H3)*.

We have conducted histone H3 ChIP and the normalization is now been included for Figures 1, 2, 3 and 4, and we
note that it has no effect on our conclusions.

*6) The conclusions from*
Figure 2
*are vague and not supported by the data presented. The authors state that “the
presence or absence of the 3Me-H3K4 mark is an important part of the chromatin
signature orchestrated by REST”. Deletion of Rest affected the H3K4me3 mark of each
set of genes in all possible ways*.

We apologize for the confusion. Our intent was to demonstrate that there is no
correlation between the changes observed in H3K27me3 levels (increased, decreased, or
unchanged) and the changes in H3K4me3. As the reviewer noticed, the deletion of REST
affected the H3K4me3 mark in these H3K27me3-defined classes in all possible ways,
indicating that H3K27me3 has no effect on H3K4me3 levels at REST sites. In addition, we
have now included more gene promoters in our analysis of H3K4me3 levels after REST
deletion (new Figure 4) and re-analyzed the
data. A majority of them gain H3K4me3 (at least 1.5-fold) after loss of REST. We
interpret this result to suggest that H3K4me3 is an important part of the chromatin
signature of REST.

*7) Similarly, for data presented in*
Figure 3*, Rest deletion
affected gene expression of ESCs, EBs and neuronal precursors in all possible ways. I
couldn't find a common trend in any set of genes*.

The reviewer is exactly right, that there was no trend. We had tried to indicate that
the results showed no correlation between changes in H3K27me3 levels and changes in gene
expression due to loss of REST. We have now replaced this data with an RNA-seq analysis
at the ESC stage showing more clearly this lack of correlation in Figure 3.

*The authors do not describe properly the results obtained or the model system
they used. For a reader not familiar with the ESC differentiation protocols, it would
be impossible to understand the rationale behind these experiments. In sum, there is
no difference between the 4 groups of genes analysed in*
Figure 3.

The reviewer arrived at the correct conclusion, namely that changes in gene expression
are not dependent on changes to H3K27me3. To simplify and underscore this result in the
resubmission, which will be unexpected to some, we have decided to focus solely on
chromatin and gene expression changes at the ESC stage. Indeed, the changes that occur
during neuronal differentiation in culture are not likely representative of the
*in vivo* conditions, given the heterogeneity in progenitors and
neuronal cell types. For this reason, the differentiation changes might best be studied
by single cell transcriptome analysis, which is outside the scope of this study.

*If the authors' only conclusion is that deletion of the Rest repressor leads to
an increase H3K4me3, this is in my opinion already well-demonstrated. In any case,
the author should comment on the fact that there are no differences in expression of
the genes analysed in*
Figure 3
*between wild-type and Rest-/- mature neurons, which is quite
surprising*.

The lack of difference in expression of REST target genes in fully differentiated
neurons is not surprising because REST is expressed poorly in these cells; it is
down-regulated at terminal differentiation to allow expression of terminal genes.
However, as we indicated above, we have now removed this data to make the work more
coherent.

*It is very far fetched to draw any general conclusion when all the analyses
performed in*
Figure 3
*are based on 12 genes. And it is unclear why the authors used only 12 of the 14
genes presented in*
Figure 3.

In response to this comment, we did a power analysis to estimate the number of genes we
should analyze, given the correlation we observe, to have a falsehood rate < 0.05.
Based on this, we extended the analysis to 22 genes and now show that the correlation
between H3K4me3 gain and increases in expression are significant (p<0.01, Figure 3).

*8) In my opinion the only interesting (novel) data in this manuscript are those
presented in*
Figure 4*: the authors
should start from here and develop this story further!*

*For example, they should perform this analysis in a genome-wide manner, with
overlapping RNA-seq analysis in wild-type and REST -/- cells in the 4 stages
(including also mature neurons)*.

Please see response to point 7 above. We have significantly extended the ESC analysis
using RNA-seq, but have eliminated the neuronal differentiation aspect for clarity and
in order to focus on PRC2 regulation in stem cells specifically, as the *in
vitro* differentiation may not accurately reflect the *in
vivo* situation.

In regards to the novelty issue, perhaps we obscured some of the novelty for this
reviewer in the previous submission. We believe that there is substantial novelty in
this work from the point of view of testing the idea, firm in the literature, that
Polycomb bivalency is a good proxy for identifying poised genes in all lineages in
pluripotent embryonic stem cells, and that PRC2 is required for poising of neuronal
genes regulated by REST. In addition, many studies would suggest that simply the
presence of polycomb proteins on a site infers repression, and for that reason other
repressor mechanisms have been ignored. There are not many repressors that could provide
good tests of the bivalent model because the complete set of targets have not been
identified, and so it is not easy to discriminate functional significance of the
binding. REST is unique: it binds to a unique sequence that allows identity of the
binding sites both bioinformatically and by ChIP; the gene targets represent a large set
of proteins that are essential for terminal neuronal differentiation, a specific
transition during differentiation; and REST is present in ESCs, where these targets are
repressed, and minimally expressed at terminal differentiation. Our study shows that
Polycomb bivalency is not a general mechanism for poising genes: rather, poising may
simply be an active repression mechanism conducted by any repressor (exemplified in this
study by REST) balancing RNA Pol II activity, indicated by the H3K4me3 mark. Further, in
the neuronal lineage, PRC2 represses proneural genes ([37], Mol. Cell, and Burgold et al, 2008, PLoS One), which are
not REST targets as identified in our REST ChIP-seq analysis, while REST mediates
repression of the later caste of genes required for terminal differentiation, which is a
quite considerable number. Our study also demonstrates that in ESCs, the presence of the
H3K27me3 mark does not necessarily equate with PRC2-based repression, as we find
de-repression after loss of REST even at REST targets that also contain the H3K27me3
mark (e.g. of Calb1 and Glra1 genes). This is an important point and may explain the
lack of de-repression observed at some H3K27me3-marked genes in PRC2 mutant cells (Shen
et al, 2008, Mol. Cell). We hope the reviewer is more convinced of the novelty component
in the new submission.

*9) Once again, the conclusions of Figure 5 are vague and not supported by the
authors' data: “These results suggest that REST repression in ESCs near a TSS is
mediated primarily by recruited HDACs that serve as a counterbalance to basal RNA
polymerase II activity, and that there is cross talk between HDACs and H3K4
trimethylation”*.

*The effect of TSA could be completely independent of REST. The correlation with
H3K4me3 has nothing to do with REST. This figure only shows correlation between
H3K4me3 and histone acetylation*.

We address this concern in several ways. First, we now show a strong correlation between
the changes in H3K4me3 due to loss of REST and those due to TSA addition at the specific
promoters we have analyzed (new Figure 4). This
correlation suggests that TSA treatment is affecting H3K4me3 levels similarly to loss of
REST. Second, we found that a microarray published previously treating ESCs with TSA
shows de-repression specifically at REST target genes (new Figure 4), albeit at lower levels than from loss of REST. Finally,
we show that for the genes we analyzed, the change in expression due to TSA treatment
was highly correlated with the change in expression due to loss of REST (new Figure 4). While none of these data prove a direct
link between REST, histone deacetylases, H3K4me3, and expression, they all strongly
support the model whereby the REST/HDAC complex antagonizes the H3K4me3 mark and
subsequent transcription. We have rewritten the Results and Discussion to reflect this
idea.

*10) The model presented in Figure 5D is wrong. Sox2 is not “bivalent” in ESCs,
it is actually expressed. While is not expressed in neurons*.

The present focus on ESCs necessitated eliminating this figure.